# Periostin interaction with discoidin domain receptor-1 (DDR1) promotes cartilage degeneration

**Tianzhen Han, Paolo Mignatti, Steven B. Abramson, Mukundan Attur** [ID] *

Division of Rheumatology, Department of Medicine, NYU Grossman School of Medicine, NYU Langone Orthopedic Hospital, New York, NY, United States of America

* mukundan.attur@nyulangone.org

## Abstract

Osteoarthritis (OA) is characterized by progressive loss of articular cartilage accompanied by the new bone formation and, often, a synovial proliferation that culminates in pain, loss of joint function, and disability. However, the cellular and molecular mechanisms of OA progression and the relative contributions of cartilage, bone, and synovium remain unclear. We recently found that the extracellular matrix (ECM) protein periostin (Postn, or osteoblast-specific factor, OSF-2) is expressed at high levels in human OA cartilage. Multiple groups have also reported elevated expression of Postn in several rodent models of OA. We have previously reported that in vitro Postn promotes collagen and proteoglycan degradation in human chondrocytes through AKT/β-catenin signaling and downstream activation of MMP-13 and ADAMTS4 expression. Here we show that Postn induces collagen and proteoglycan degradation in cartilage by signaling through discoidin domain receptor-1 (DDR1), a receptor tyrosine kinase. The genetic deficiency or pharmacological inhibition of DDR1 in mouse chondrocytes blocks Postn-induced MMP-13 expression. These data show that Postn is signaling though DDR1 is mechanistically involved in OA pathophysiology. Specific inhibitors of DDR1 may provide therapeutic opportunities to treat OA.

## Introduction

Osteoarthritis (OA) affects more than 20% of the population over age 60 and is a significant cause of disability for millions of people. The precise cellular mechanisms that drive OA progression are not fully understood; however, it is clear that events in all three compartments–cartilage, bone, and synovium–play important roles in the initiation and progression of OA. The analysis of the molecular mechanisms underlying dysregulation of cartilage homeostasis and alteration of subchondral bone in OA can provide important insights, and lead to novel strategies for disease-modifying treatments and joint repair. Periostin (Postn), a TGFβ-inducible, ECM protein, also known as Osteoblast-Specific Factor 2 (OSF2), is expressed in multiple joint compartments (cartilage, subchondral bone, meniscus, ligaments, and osteophytes) [1–5]. Our previous studies have shown that Postn is overexpressed in human OA cartilage and

**Data Availability Statement:** All relevant data are within the paper and its Supporting Information files.

**Funding:** Funding for this project provided by the National Institute of Health grant AR054817,

AR068341, and AR070934. The funders had no role in study design, data collection, and analysis, decision to publish, or preparation of the manuscript

**Competing interests:** The authors have declared that no competing interests exist.

surgical models of OA in rodents (MM, ACL, and ACL/PM) [1]. These findings suggest that Postn, first reported by us to be upregulated in human OA, plays a role in the pathogenesis of OA, where it exerts catabolic effects in cartilage. Furthermore, Postn also plays a role in tissue repair. Insufficient levels of Postn result in impaired tissue remodeling [6], whereas elevated expression of Postn is associated with chronic diseases such as liver fibrosis, atherosclerotic and rheumatic cardiac valve degeneration, as well as OA [7–9].

The nature of Postn receptor(s) in chondrocytes is unknown. A recent study reported that Postn interacts with discoidin domain receptor 1 (DDR1), a collagen-binding receptor in myofibroblasts, and promotes lysyl oxidase expression [8, 10]. DDR1, a type I transmembrane protein with a cytoplasmic C-terminal tyrosine kinase domain, undergoes ligand-induced autophosphorylation, with maximal activation (phosphorylation) achieved hours after ligand binding [11]. DDR1 and the closely related DDR2 are expressed in chondrocytes, and both DDR1 and DDR2 bind collagen [12]. Upon collagen-binding, DDR1 dimerizes, with autophosphorylation of tyrosine-residues and activation of downstream signaling pathways, including p38, ERK1/2, and JNK MAP kinases, as well as PI3 kinase-AKT [13, 14] and Wnt/β-catenin signaling [15]. Since DDR1 is a putative binding partner of Postn and can activate inflammatory signaling pathways [16–18], we investigated the involvement of the Postn—DDR1 interaction in cartilage degeneration.

## Materials and methods

### Reagents

All media and FBS were purchased from Thermo Fisher Scientific (Waltham, MA); proMMP-13 enzyme-linked immunosorbent assay (ELISA) kits from R&D systems (Minneapolis, MN); Phosphatase Inhibitor Cocktail [#5870; Cell signaling, Danvers, MA, USA] and protease inhibitor cocktail (#539131, Calbiochem, San Diego, CA, USA); The Wnt signaling inhibitor CCT031374 hydrobromide (cat. no. 4675/10, R&D Systems, Minneapolis, MN), antibodies to β-Catenin (D10A8; XP® Rabbit mAb #8480), α-Tubulin (11H10; Rabbit mAb#2125), Akt (#9272); Phospho-Akt (Ser473) (193H12; Rabbit mAb #4058), Phospho-DDR1 (Tyr792; #11994) and total DDR1 (D1G6; XP® Rabbit mAb #5583) from Cell Signaling Technology (Danvers, MA 01923); the antibodies to human Phospho-DDR2 (Y740; #1119D) and human DDR1 (AF2538) from R&D system (Minneapolis, MN); HRP-conjugated rabbit anti-goat (#305-036-047), HRP-conjugated goat anti-rabbit (#111-035-144), AffiniPure goat anti-human lgG, F(ab')2 fragment specific (#109-005-097) and peroxidase affinipure goat anti-human lgG(H+L; #109-035-088) from Jackson ImmunoResearch (West Grove, PA, USA). Human DDR1-full length [19] and DDR1-Fc [20] were kind gifts from Dr. Friedemann Kiefer, Max Planck Institute for Molecular Biomedicine, Germany, and from Birgit Leitinger, National Heart and Lung Institute, Imperial College London, London UK, respectively.

### Procurement of human cartilage

Tibial and femoral knee articular cartilage was obtained from patients with advanced OA (age 50–70 yrs.) undergoing knee replacement surgery at NYU Langone Orthopedic Hospital. The patients had been free of non-steroidal anti-inflammatory drugs for at least two weeks before surgery. A certified pathologist inspected the cartilage for gross morphology prior to chondrocyte isolation. All specimens exhibited large areas of thinning of the cartilage, granularity, and focal eburnation of the underlying bone. Thus, the cartilage in our studies was heterogeneous with respect to the OA disease stage. Their use in the de-identified form in the current research was in accordance with the ethical standards of the Helsinki Declaration of 1975, as revised in 2000, and was approved (#i9018) by the institutional review board (IRB) of the NYU School of

Medicine. The ethics committee waived the requirement for informed consent. The surgical discards from knee replacement surgery were obtained for human chondrocyte isolation and cartilage for histology and immunohistochemistry (IHC) studies.

### *Ddr1* knock out mice

$Ddr1^{-/-}$ mice were kindly provided by Dr. Edward Skolnik [21] (NYU School of Medicine). $Ddr1^{-/-}$ and wt mice were maintained on a C57BL6 background, and heterozygote ($Ddr1^{+/-}$) males and females were bred to produce wild type (wt) and Ddr1 knockout ($Ddr1^{-/-}$) littermates for analysis. The mice were fed standard pelleted chow and allowed access to water ad libitum. Mice were euthanized by $CO_2$ inhalation and cervical dislocation, and joint tissues were harvested at 10–12 weeks for chondrocyte isolation. All animals were used in accordance with scientific, humane, and ethical principles and compliance with regulations approved by the New York University School of Medicine Institutional Animal Care and Use Committee (IACUC). Genotyping performed using genomic DNA isolated from mouse tail biopsies using the following primers: Primer 1, GCAGCGCATCGCCTTCTATC, and primer 2, AGACAATCTC GAGATGCTGG, to amplify the wild-type allele, generating a 250-bp PCR product; Primer 3, GTTGCGTTACTCCCGAGATG and primer 1 to identify the mutant allele, amplifying a 160-bp PCR product.

### OA chondrocyte culture

Human OA chondrocytes were harvested from discarded end-stage knee OA cartilage as described [1]. Briefly, cartilage slices were minced finely and digested with 0.2% collagenase for 12–16 h in Ham's F12 medium supplemented with 5% FBS. The resulting cell suspension was seeded into T175 flasks and allowed to grow for 48 h. In all experiments, primary chondrocytes used between passage 1 and 2, plated at 80% confluence in 6- or 12- or 24-well plates.

### Mouse chondrocyte isolation

Cultures of murine chondrocytes were established by a modification of the method described [22, 23]. Mouse primary articular chondrocytes from 10-12-week old mice were isolated from four mice for each genotype (approximately $5 \times 10^6$ cells) and were grown in DMEM containing 10% FBS and antibiotics. Passage-1 or -2 chondrocytes were used in all the experiments. The experiments were repeated at least twice, with cells isolated from different sets of animals. We also confirmed the chondrocyte phenotype by determining type I and II collagen and aggrecan expression, as well as the ratio between type II and I collagen levels by qRT-PCR. Cells were grown in 12–well plates were also stained with Alcian blue to confirm the chondrocyte phenotype.

### Measurement of proMMP-13 protein and activity

Secreted MMP-13 levels were measured in the cell-conditioned medium of monolayer cultures by an ELISA kit from R&D Systems (Minneapolis, MN). To measure MMP-13 activity, the culture supernatant was incubated with Assay Buffer [50 mM Tris.Cl (pH7.5), 10 mM Cacl2, 150 mM NaCl, 0.05% Brij35] and APMA (p-aminophenylmercuric acetate) for two hours at $37^o$ C. APMA-activated samples (20 μL) were incubated with the MMP-13-specific fluorogenic substrate, MOCAc-Pro-Cha-Gly-Nva-His-Ala-Dap (DNP)-NH2 (Peptide International, Louisville, KY) at $37^o$ C for one hour, and the fluorescence was measured with a Synergy HT microplate reader at 360/40 nm (excitation) and 460/40 nm (emission). Recombinant pro-MMP-13 (R&D) was used as a positive control and for generating a standard curve.

## Transient transfection

C28/I2 chondrocytes (a human chondrocyte cell line kindly provided by Dr. Mary B. Goldring, Hospital for Special Surgery, New York, NY, USA) were seeded in 10-cm dishes at 80% confluency in DMEM supplemented with 10% FBS, and transfected with 10 μg of the full-length Postn and/or DDR1-Fc cDNAs described under Reagents, using the TransIT-LT1 transfection reagent (Mirus Bio, Madison, WI, USA) in Opti-MEM. After 24 h, the medium was changed to serum-free DMEM. Twenty-four to 48 h post-transfection, the cells were lysed for immunoprecipitation (IP).

## Adenovirus transduction

Mouse or human chondrocytes seeded at 70% confluency were grown for 24 h in DMEM supplemented with 10% FBS. After replacing the medium with fresh medium, the cells were transduced with control, empty adenovirus (ad-CMV-Null)#1300) or a mouse Postn-encoding adenovirus (Ad-m-Postn; #ADV-269083; (Vector Biolabs, Malvern, PA, USA) at MOI = 100 for four hours, after which the medium was changed to serum-free medium. Twenty-four-hours post-transduction, the cells, and supernatants were collected for the MMP-13 activity assay, and the cells lysed in TRIzol for RNA extraction.

## Western blotting and immunoprecipitation

Cells were lysed with RIPA buffer containing a phosphatase and protease inhibitor cocktail. Cell protein (30–40 μg) was used for Western blotting analysis as described [1, 24]. C28/I2 cells seeded at 80% confluency were grown overnight in DMEM supplemented with 10% FBS, and then starved for 24 h in serum-free medium. The cells were then transfected with DDR1-Fc and/or Postn cDNA as described [1]. After 24–48 h, the cells were washed with cold PBS and lysed with RIPA buffer containing phosphatase and protease inhibitors. One milligram of cell extract protein was mixed with 10 μl Protein A & G Sepharose (Sigma-Aldrich) and 10 μl anti-Postn (Sigma-Aldrich) or anti-DDR1 antibodies (Cell Signaling), or with control IgG for 24 h at 4˚ C. Antigen-antibody complexes were then analyzed by Western blotting as described above.

## RNA extraction and qRT-PCR analysis

Total RNA was extracted from chondrocyte cultures using TRIzol, precipitated with isopropanol, and further purified using the micro RNeasy column cleanup kit (Qiagen) following the manufacturer's instructions. One microgram of RNA was used for cDNA synthesis using the Advantage℞ RT-for-PCR Kit (Takara Bio USA, Inc. Mountain View, CA). Pre-designed TaqMan primer sets were purchased from Thermo Fisher Scientific (Waltham, MA USA). Real-time PCR was run in an Applied Biosystems Prism 7300 sequence detection system. Messenger RNA levels were normalized to GAPDH and 18S mRNA, and the relative expression levels of various transcripts were calculated using an approximation method or the 2 delta CT method [25].

## Densitometry

Quantitative analysis of Western blot bands was performed with ImageJ 1.52a software (National Institutes of Health). The results are presented as the fold change between the reading of the sample and that of the corresponding loading control.

## Statistical analysis

All the experiments were done with three to four chondrocyte cultures from individual OA patients, and each sample was assayed in duplicate or triplicate. The data are expressed as mean ± standard deviation (m ± SD). The non-parametric Mann-Whitney test was used to calculate p values. For the analysis of the experiments with AdCon or AdPostn and the DDR1 inhibitor 7rH, one-way ANOVA was performed with multiple testing using Prism 7 (Graph-Pad). P values <0.05 were considered statistically significant.

## Ethics statement

This study was carried out in strict accordance with the recommendations in the Guide for the Care and Use of Laboratory Animals of the National Institutes of Health. The Institutional Animal Care and Use Committee (IACUC) of the New York University School of Medicine under protocol number IA16-00601 approved all procedures performed in the study.

# Results

## Expression of DDR1 in human OA and normal cartilage

We first analyzed DDR1 expression in human OA and normal cartilage. As shown in Fig 1, DDR1 mRNA expression was comparable in OA and normal cartilage by microarray (Fig 1A). These data were further validated by qPCR (ct values 26.66 and 26.74 respectively) in different sets of human OA and normal cartilage samples (Fig 1B), and also by immunohistochemistry (Fig 1C) of lesional (damaged) and non-lesional areas (preserved) of the cartilage from the same patient. Thus, DDR1 is not differentially expressed in OA relative to age-matched, normal, or preserved cartilage.

## DDR1—Postn interaction

To investigate whether Postn binds to DDR1 in chondrocytes, we performed co-immunoprecipitation (co-IP) experiments with anti-Postn and DDR1 antibodies using primary OA chondrocytes. As shown in Fig 2, DDR1 and Postn co-immunoprecipitate, showing the presence of DDR1—Postn complexes in human primary chondrocytes.

In primary chondrocytes, the level of Postn expression was too low relative to DDR1. Therefore, we took an alternative approach to verify the Postn—DDR1 interaction using the human chondrocyte cell line C28/I2. We transfected C28/I2 cells with a pDDR1-Fc fusion protein construct and with a plasmid expressing human pPostn and analyzed them by IP 24–48 h later. As shown in Fig 2, the results of these co-IP experiments showed that Postn forms complexes with DDR1 but not with DDR2.

## DDR1 inhibition abrogates postn induction of MMP-13 expression

We next examined the effect of DDR1 inhibition on Postn induction of MMP-13 expression. As shown in Fig 3, human primary OA chondrocytes constitutively express MMP-13 mRNA and protein, which were strongly upregulated following transduction with adenovirus encoding human Postn (Ad-hPostn). The addition of the synthetic DDR1 inhibitor 7rh to the culture medium dose-dependently inhibited this effect. As expected, based on our previous results [1], Wnt signaling inhibition significantly blocked Ad-hPostn-induced expression of MMP-13. We next investigated the effect of Postn on DDR1 and DDR2 phosphorylation. As shown in Fig 4A, Postn induced DDR1 but not DDR2 phosphorylation without affecting the total levels of either protein.

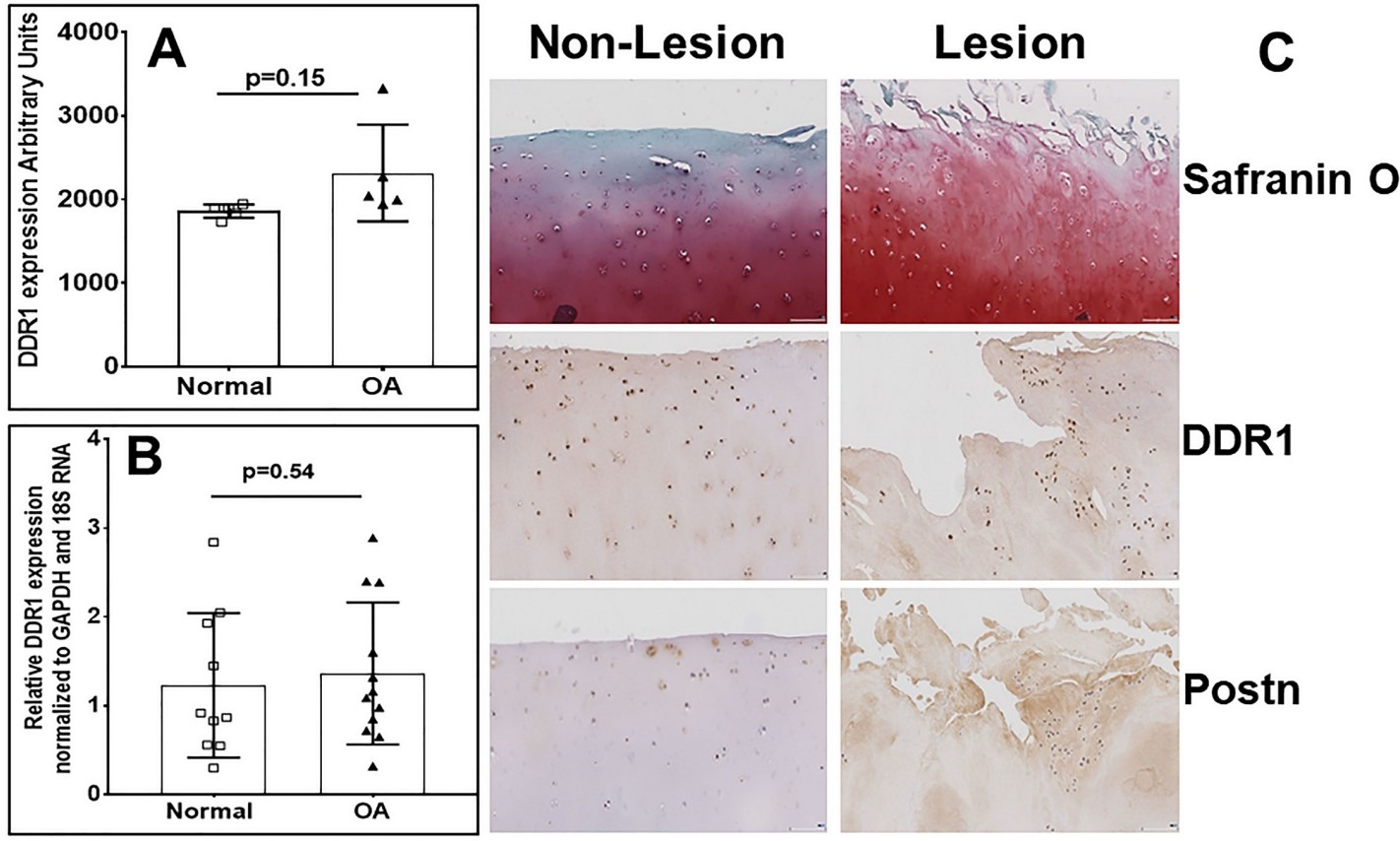

**Fig 1. Elevated expression of Postn but not DDR1 in OA cartilage. A)** Gene array analysis. Total RNA was isolated from six pools of normal and OA cartilage. For each pool, 5–10 individual cartilage samples (1 g each) were pooled. RNA was hybridized against human U133A arrays. After normalization, DDR1 expression was expressed in arbitrary units. **B)** qPCR analysis of DDR1 mRNA in 10 normal and 12 OA cartilage specimens from human knee joints. Mean (SD) are shown. The non-parametric Mann-Whitney test determined statistical significance. **C)** Immunohistochemical analysis of lesional (damaged) and non-lesional (preserved) OA knee cartilage shows cell- and ECM-associated Postn in lesional areas, whereas DDR1 is cell-associated. OA knee cartilage from a total knee replacement was sectioned and stained for Postn and DDR1 with Vectastain reagents (Vector Laboratories, Burlingame, CA, USA).

### DDR1 –ß-catenin mediates postn signaling in chondrocytes

We have previously shown that inhibition of the Wnt signaling pathway blocks the Postn-induction of MMP-13 expression in OA chondrocytes [1]. Therefore, we investigated whether DDR1- Postn interaction activates AKT-ß-catenin signaling in human OA chondrocytes. Western blotting analysis of phosphorylated DDR1 and AKT, and of total β-catenin showed that treatment of human OA chondrocytes with Postn (5 μg/ml) induced DDR1 and AKT phosphorylation and increased β-catenin levels in a time-dependent manner (Fig 4B). Furthermore, the DDR1 inhibitor 7rh blocked Postn induction of AKT phosphorylation and decreased β-catenin level (Fig 4C), showing that Postn activation of DDR1 leads to β-catenin signaling and MMP-13 expression in human chondrocytes.

Since Postn interaction with αvβ3 and αvβ5 integrins activates downstream signaling in bone and other cells [26–28], we also tested the effect of neutralizing antibodies to αvβ3 (LM609) or αvβ5 (P1F6) on chondrocyte expression of MMP-13. Neither antibody blocked Postn induction of MMP-13 expression or activity (data not shown).

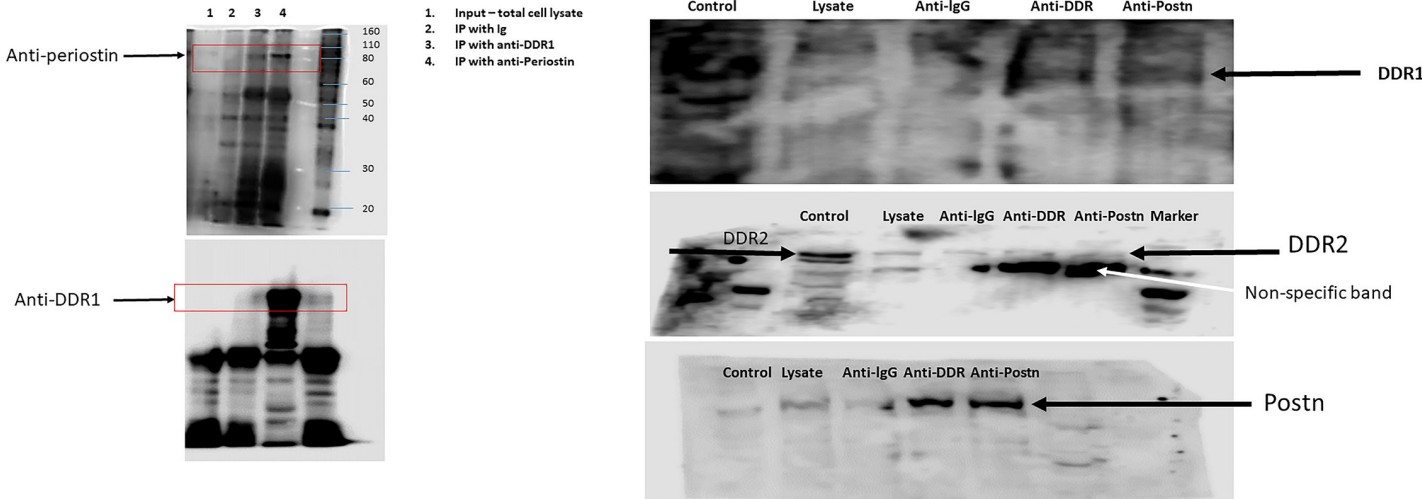

**Fig 2. Co-immunoprecipitation of DDR1-Postn complexes in human chondrocytes. A**: Western blotting analysis of immunoprecipitates from primary human OA chondrocytes. No antigens are seen in the non-immunoprecipitated cell lysate (lysate) because their concentration is too low. **B:** Co-immunoprecipitation of DDR1/Postn complexes in human chondrocyte C28/I2 cells. A representative Western blot of immunoprecipitates from chondrocytes is shown. Purified DDR1 and DDR2 proteins were loaded as a positive control.

## Postn dost not induce MMP-13 expression in *Ddr1* deficient chondrocytes

To investigate whether receptors other than DDR1 can mediate Postn induction of MMP-13 expression, we examined the effect of Postn on MMP-13 expression in *Ddr1*-deficient mouse chondrocytes.

For this purpose, we isolated chondrocytes from *Ddr1*$^{-/-}$ and *Ddr1*$^{+/+}$ (wt) mice. Adenovirus-mediated overexpression of mouse Postn (Ad-mPostn; Vector Labs) increased MMP-13 expression (Fig 5A), and activity (Fig 5B) in wt chondrocytes but had no such effect in *Ddr1*$^{-/-}$

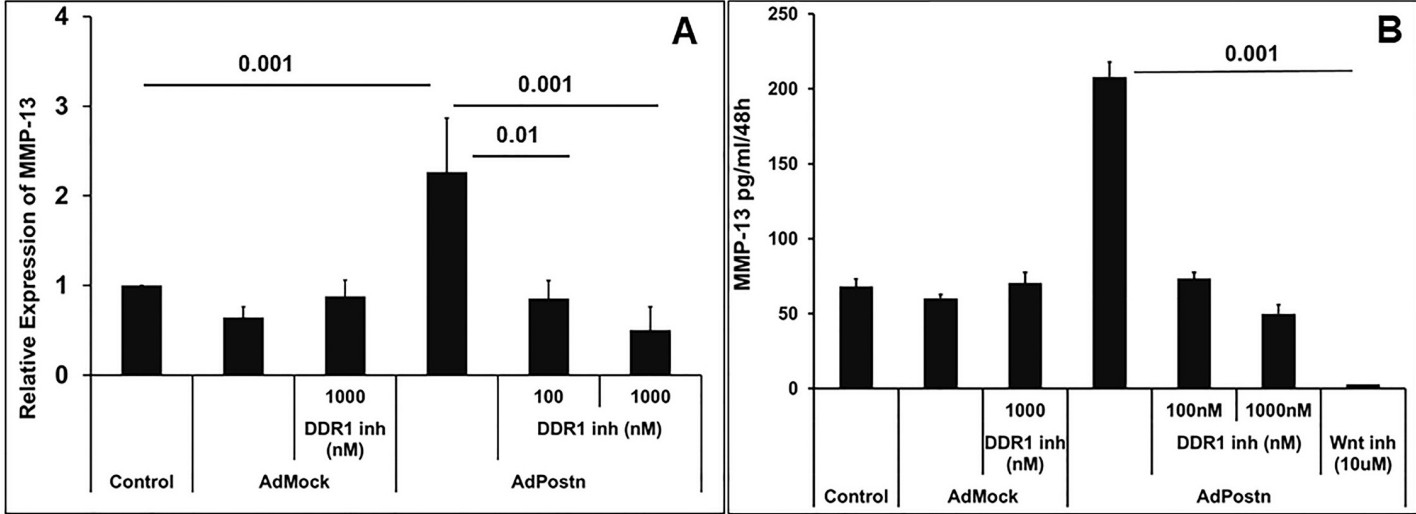

**Fig 3. The DDR1 inhibitor 7rh inhibits the Adenovirus-mediated induction of MMP-13 expression of human Postn (Ad-hPostn) in human OA chondrocytes.** Human OA chondrocyte cultures were pre-incubated in the presence or absence of DDR1 inhibitor (7rh; 100–1000 nM) for 16 h before Ad-hPostn or AdCon virus (MOI = 100) was added to the cultures. Total RNA was isolated 24 h post-transduction to measure MMP-13, and GAPDH expression (A). The supernatant was used to determine MMP -13 levels by ELISA (. Mean ± SD of triplicate samples are shown. 1-way ANOVA determined significance with the post-Bonferroni multiple comparison test. P <0.05 was considered significant. DDR1 and wnt signaling inhibition reduced MMP-13 expression.

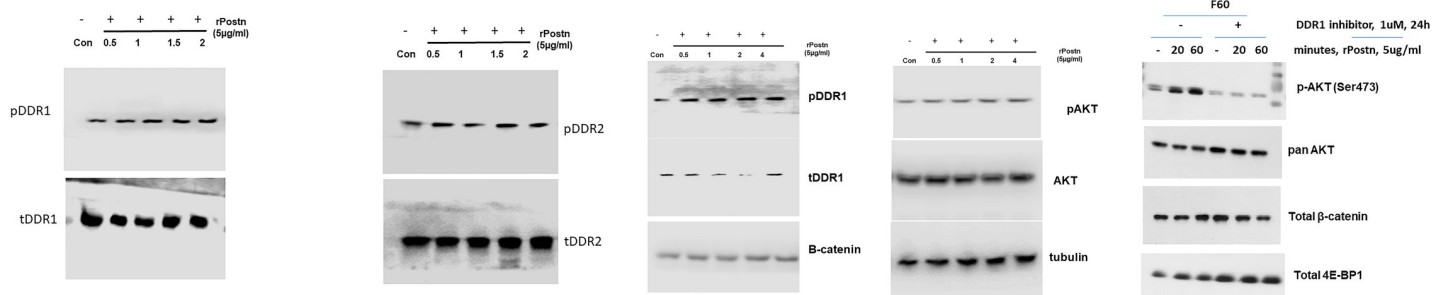

**Fig 4. DDR1 kinase mediates Postn activation of AKT and β-catenin signaling in human chondrocytes: A: Postn induces DDR1 but not DDR2 phosphorylation. Upper panel:** Western blotting analysis of phosphorylated (p) and total (t) DDR1 and DDR2 in human chondrocytes treated with Postn (5 μg/ml) for the indicated times. Total DDR1 and DDR2, are shown as loading controls. A representative Western blot is shown. Lower panel: densitometric analysis of two similar Western blots. The columns represent the densitometric readings of the indicated bands normalized to those of the corresponding loading controls. **B: Postn activates AKT and β-catenin signaling.** Upper panel: Western blotting analysis of AKT and β-catenin activation in human articular chondrocytes treated with Postn (5 μg/ml) for the indicated times. A representative Western blot of two independent primary human chondrocyte cultures is shown. Lower panel: densitometric analysis of two similar Western blots. The columns represent the densitometric readings of the indicated bands normalized to those of the corresponding loading controls. **C: Inhibition of DDR1 kinase blocks Postn induction of AKT and β-catenin signaling.** Western blotting analysis of AKT phosphorylation (pAKT473) and β-catenin levels in human articular chondrocytes treated with Postn (5 μg/ml) in the presence or absence of the DDR1 inhibitor 7rh (1 μM) for the indicated times. Pan AKT and 4E-BP1 are shown as loading controls. A representative blot of two independent chondrocyte cultures is shown. *p<0.05 versus control.

chondrocytes. However, expression of exogenous human DDR1 (hDDR1) rescued MMP-13 expression in Ddr1$^{-/-}$ chondrocytes (Fig 5B). In hDDR1-transfected wt cells, MMP-13 activity increased in both control and Ad-mPostn transduced cells. Conversely, Ad-mPostn transduction had no such effect in Ddr1$^{-/-}$ cells. Thus, consistent with our previous results (Fig 3), these results showed that Postn-dependent MMP-13 expression requires signaling via DDR1.

## Discussion

Our results provide strong evidence that the effects of Postn on MMP-13 expression require interaction with DDR1 and activation of the Wnt/β-catenin signaling pathway. Several lines of evidence support this conclusion. First, co-immunoprecipitation experiments show Postn

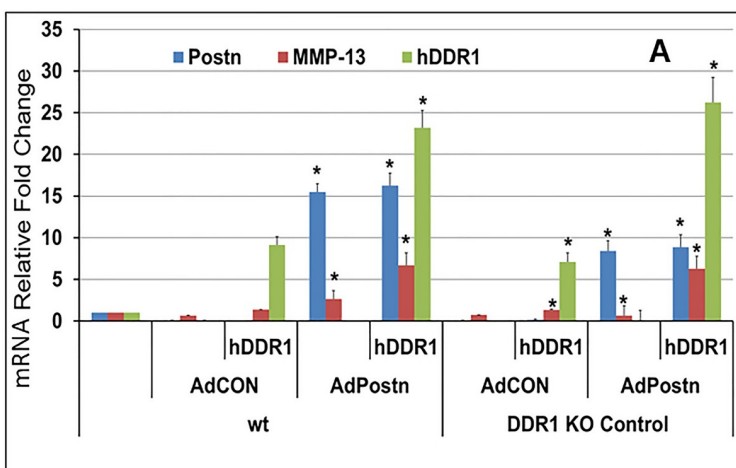

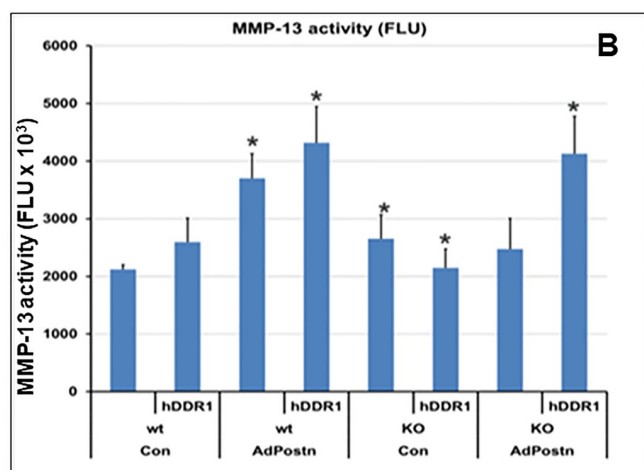

**Fig 5. DDR1 mediates Postn induction of MMP-13 expression in mouse chondrocytes.** Mouse chondrocytes were isolated from three *wt and Ddr1* deficient mice, as described in Materials and Methods. Cells were grown in 12-well plates at a density of 5x10$^4$ cells/well. Chondrocytes were transfected with human DDR1 (hDDR1) and also transduced with AdCon or Ad-mPostn in triplicate. Cells were lysed 24 h post- adenovirus transduction in TRIzol for RNA isolation, and the supernatant was collected for MMP-13 activity as described in methods. RNA expression was normalized to GAPDH and expressed as relative fold-change to wt non-transfected and -transduced cells. Mean ± SD of triplicate samples are shown. *: p<0.05 by 1-way ANOVA; vs. wt hDDR1 transfected or AdCon-transduced cells.

association with DDR1, but not with DDR2. Second, exposure of human chondrocytes to Postn induces DDR1, but not DDR2, phosphorylation. Third, periostin–DDR1 interaction results in AKT activation and increased β-catenin levels. We have previously reported that Postn activates Wnt/β-catenin signaling, and the Wnt signaling inhibitor CCT031374 hydro-bromide, which blocks β-catenin stabilization and thus inhibits TCF-dependent transcription, abrogates Postn induction of MMP-13 expression in human chondrocytes [29]. In the present study, the DDR1 inhibitor 7rh also blocked Postn-induced MMP-13 upregulation, as well as AKT phosphorylation and β-catenin stabilization. Finally, Postn did not induce MMP-13 expression in $Ddr1^{-/-}$ chondrocytes.

These findings show that Postn–DDR1 signaling is mediated by β-catenin stabilization, and therefore by canonical Wnt signaling. We propose that activation of β-catenin stabilization is mediated by DDR1 activation of the PI3K-AKT pathway. AKT can stabilize β-catenin by two mechanisms: phosphorylation of β-catenin at $Ser^{552}$ [30] and inhibition of glycogen synthase kinase-3 (GSK3) [31], which phosphorylates β-catenin, causing its ubiquitination and degradation. Another potential mechanism could consist of AKT upregulation of canonical Wnt growth factors; however, we could not find any such report in the literature.

What is the relevance of these findings to the pathogenesis of OA? Postn has the unique property of stimulating catabolic effects in cartilage while promoting osteogenesis in bone. Postn is produced by multiple joint tissues, including bone, cartilage, ligaments, meniscus, and synovium under normal or injury-induced conditions [29]. Karlsson et al. and two other groups have independently shown increased expression of Postn in cartilage and subchondral bone of human OA-affected joints [3, 4, 32–34]; and elevated levels of Postn have also been reported in human OA synovial fluids, as well as in synovial fluids of anterior cruciate ligament injury patients [35–37]. Loeser et al. have reported that Postn expression is elevated in murine knee cartilage and menisci with surgically-induced OA [5], and elevated levels of Postn have also been described in the anterior cruciate ligament (ACL), as well as in murine models of surgically-induced post-traumatic OA [1–3, 5, 33, 38, 39]. Furthermore, plasma Postn levels are markedly elevated in post-menopausal women with non-vertebral fractures, even after adjustment for bone mineral density (BMD) [40]. Recent studies also reported that Postn levels in synovial fluid correlate with radiographic signs of knee OA [41], and that Postn is the only protein whose expression levels significantly correlate with OARSI score, synovitis, and microCT-based bone parameters (e.g., Structure Model Index and trabecular threshold). Postn influences ECM homeostasis, bone remodeling, and OA development [42]. Therefore, Postn offers a unique target for OA treatment since it can mediate pathogenic events in both cartilage and bone. Furthermore, we have previously shown that Postn signals via the AKT-Wnt signaling pathway to induce MMP-13 and ADAMTS 4 expression [1]. Postn inter-acts with integrins αvβ3 and αvβ5 to activate downstream signaling in bone and other cells [26–28, 43, 44]. Postn also promotes macrophage proliferation and polarization [45]. Our pre-liminary qRT-PCR analysis has shown that DDR1 is expressed in normal human and mouse chondrocytes and is not differentially expressed in OA cartilage. The DDR1 inhibitor 7rh blocks constitutive and Postn-induced MMP-13 expression in human chondrocytes with a dose-dependent effect. This inhibitor has 3–4 fold selectivity for DDR1 over DDR2 [46, 47] (currently no DDR2-specific inhibitor is available).

In our in vitro experiments with human chondrocytes, the addition of Postn to the culture medium induced DDR1, but not DDR2, phosphorylation, and the DDR1 inhibitor 7rh blocked AKT phosphorylation and β-catenin stabilization. $Ddr1^{-/-}$ mice were recently found to spontaneously develop osteoarthritis of the temporomandibular joint, suggesting that loss of $Ddr1$ and associated increases in Ddr2 may initiate MMP-13 upregulation, resulting in type 2-collagen degradation [48]. However, in our in vitro studies, we did not observe increased

Ddr2 expression in *Ddr1* deficient chondrocytes (data not shown). Whether Postn-DDR1 interaction also promotes catabolic events in other joint tissues, including synovial proliferation, remains an open question. Postn has been shown to promote liver and adipose tissue inflammation and fibrosis [8, 49].

In conclusion, our data shed new light on the mechanism of Postn induction of catabolic effects in knee joint cartilage. MMP-13 upregulation by Postn requires interaction with and activation of DDR1, which signals via the AKT-Wnt/ β-catenin pathway. Inhibition of Postn signaling through DDR1, therefore, could provide therapeutic opportunities for OA.

## Supporting information

**S1 Fig.**
(PPTX)

## Author Contributions

**Conceptualization:** Tianzhen Han, Paolo Mignatti, Steven B. Abramson, Mukundan Attur.

**Formal analysis:** Tianzhen Han, Paolo Mignatti, Steven B. Abramson, Mukundan Attur.

**Investigation:** Tianzhen Han, Paolo Mignatti, Mukundan Attur.

**Methodology:** Tianzhen Han, Paolo Mignatti, Mukundan Attur.

**Resources:** Tianzhen Han, Mukundan Attur.

**Writing – original draft:** Paolo Mignatti, Steven B. Abramson, Mukundan Attur.

**Writing – review & editing:** Tianzhen Han, Paolo Mignatti, Steven B. Abramson, Mukundan Attur.

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
