## [Decision Letter · Decision Letter 0]

21 Jan 2020

PONE-D-19-33141

PERIOSTIN INTERACTION WITH DISCOIDIN DOMAIN RECEPTOR-1 (DDR1) PROMOTES CARTILAGE DEGENERATION

PLOS ONE

Dear Dr Attur,

Thank you for submitting your manuscript to PLOS ONE. After careful consideration, we feel that it has merit but does not fully meet PLOS ONE’s publication criteria as it currently stands. Therefore, we invite you to submit a revised version of the manuscript that addresses the points raised during the review process.

We would appreciate receiving your revised manuscript by 22 March 2020. To enhance the reproducibility of your results, we recommend that if applicable you deposit your laboratory protocols in protocols.io, where a protocol can be assigned its own identifier (DOI) such that it can be cited independently in the future. For instructions see: http://journals.plos.org/plosone/s/submission-guidelines#loc-laboratory-protocols

We look forward to receiving your revised manuscript.

Kind regards,

Dominique Heymann, Ph.D.

Academic Editor

PLOS ONE

Journal Requirements:

2.  In the ethics statement in the manuscript and in the online submission form, please provide additional information about the human cartilage used in your study. Specifically, please ensure that you have discussed whether researchers had access to potentially identifying information and/or whether the IRB or ethics committee waived the requirement for informed consent. If patients provided informed written consent to have their discarded cartilage tissue used in research, please include this information.

- In the Methods, please provide the following information about the mice used in this study: (1) Please state the genetic background of the Ddr1-/- mice; (2) Please provide details of animal welfare (e.g., shelter, food, water, environmental enrichment); (3) Please include the method of euthanasia.

"Funding for this project was provided by the National Institute of Health grant

 AR054817, AR068341, and AR070934. The funders had no role in study design, data

 collection, and analysis, decision to publish, or preparation of the manuscript."

"None"

Reviewers' comments:

Reviewer's Responses to Questions

**Comments to the Author**

1. Is the manuscript technically sound, and do the data support the conclusions?

Reviewer #1: No

Reviewer #2: Yes

Reviewer #3: Yes

2. Has the statistical analysis been performed appropriately and rigorously? 

Reviewer #1: I Don't Know

Reviewer #2: Yes

Reviewer #3: Yes

3. Have the authors made all data underlying the findings in their manuscript fully available?

Reviewer #1: No

Reviewer #2: Yes

Reviewer #3: Yes

4. Is the manuscript presented in an intelligible fashion and written in standard English?

Reviewer #1: No

Reviewer #2: Yes

Reviewer #3: Yes

5. Review Comments to the Author

Reviewer #1: In this report, authors would like to prove the interaction of DDR1 and periostin and its signal in chondrocyte. However, the interaction of DDR1 and periostin and DDR1/periostin-MMP13 signal were already proved in several years ago. It is hard to find new finding.

Here author show that MMP13 was increased when periostin overexpressed chondrocyte cell line, and it is inhibited by DDR1 inhibitor. However, biological meaning of this phenotype is not significant. In OA patient, DDR1 and periostin were not increased by q-PCR and IHC. Actually, periostin and MMP13 increase in OA was reported and recombinant periostin induces DDR1-MMP13 relation was also proved in previously.

Please show us new signaling of periostin-DDR1 or new biological meaning.

Reviewer #2: Excellent paper with solid evidences deciphering the periostin pathway in chondrocytes, and showing that the pathway is different from bone pathway.

The discussion is very well constructed, with the question raised by the spontaneous OA developped by ddr-/- mice.

Just minors punctuation errors (pg 10 line 12 .,. , and some "Figure" instead "Fig. ")

Reviewer #3: In a paper previously published in FASEB J in 2015, the authors reported high periostin levels in human and mouse OA cartilage. Gain and loss of function experiments indicated that periostin stimulated MMP13 expression and promoted cartilage degradation. In the present article, they report that periostin induces collagen and proteoglycan degradation by signaling through discoidin domain receptor-1 (DDR1).

They first show that DDR1 mRNA and protein levels are not different between non-lesion and lesion OA, and was cell-associated. In contrast periostin protein was more expressed in OA lesions and was both cell and ECM-associated. Then by performing co-immunoprecipitation, they show in OA chondrocytes and in the C28I2 cell line that DDR1 interacts with periostin. Inhibition of DDR1 with the 7rh molecule blocked the increase in MMP13 levels induced by periostin adenoviral overexpression. Moreover, authors show that periostin activates AKT phosphorylation and increased beta-catenin levels, and that these effects are prevented by inhibition of DDR1. Finally, in chondrocytes from Ddr1 deficient mice, periostin does not increase MMP13 expression and activity further suggesting that periostin signals through DDR1.

This is an interesting article with original results. The results are convincing. However, the methods used and the experimental conditions are insufficiently detailed in the materials and methods section and in the figure legends.

- Abstract line 16: 2 spaces between DDR1 and is. Several other similar errors throughout the text

- Introduction: periostin is presented as a TGFb-inducible protein. TGFb is known as an anabolic cartilage GF and periostin catabolic in the context of OA. Can authors explain this apparent opposition?

- Please indicate whether you used human chondrocytes with several passages or only freshly isolated cells.

- Please indicate in the legend of figure 3 what corresponds to A or B.

- Page 11 first sentence: “strongly upregulated” written twice

- Page 11, first paragraph, last sentence (and figure 3). It is necessary to explain why authors used a wnt inhibitor in this experiment. They also have to explain which Wnt inhibitor was used, at which concentration, and discuss the results (MMP13 levels completely dropped). Is this Wnt inhibitor cytotoxic at the concentration used?

- Page 11 and Figure 4A: authors should perform densitometric quantification of the bands from several independent experiments and calculate the mean to better analyze DDR1 phosphorylation in response to periostin.

- Page 11 figure 4B: same remark. The authors have to perform densitometric analyses to fully convince the readers.

- Page 11 and page 13: authors should explain the molecular link between Akt and b-catenin. Is this activation of b-catenin independent from the canonical Wnt signaling pathway activated by Wnt growth factors?

6. PLOS authors have the option to publish the peer review history of their article (what does this mean?). If published, this will include your full peer review and any attached files.

Reviewer #1: No

Reviewer #2: No

Reviewer #3: No

---

## [Author Response · Author response to Decision Letter 0]

11 Mar 2020

Dr. Dominique Heymann, Ph.D.

Academic Editor

PLOS ONE

Sub: Submission of revised manuscript - PONE-D-19-33141

Dear Dr. Heymann

Thank you very much for your email of Jan-21-2020 with the reviews of our manuscript “PERIOSTIN INTERACTION WITH DISCOIDIN DOMAIN RECEPTOR-1 (DDR1) PROMOTES CARTILAGE DEGENERATION” (PONE-D-19-33141).

We are very grateful for your decision to allow us to resubmit our revised manuscript. We appreciate that Reviewers 2 and 3 agree with importance our findings on the role of periostin in osteoarthritis. 

Please find below our point-by-point responses to the editorial and reviewers’ comments, with detailed indication of the changes we made.

Journal Requirements:

2. In the ethics statement in the manuscript and in the online submission form, please provide additional information about the human cartilage used in your study. Specifically, please ensure that you have discussed whether researchers had access to potentially identifying information and/or whether the IRB or ethics committee waived the requirement for informed consent. If patients provided informed written consent to have their discarded cartilage tissue used in research, please include this information.

Response: OA cartilage samples were collected from end-stage knee OA patients undergoing knee replacement surgery at NYU Langone Orthopedic Hospital. Their use in de-identified form in the current study was in accordance with the ethical standards of the Helsinki Declaration of 1975, as revised in 2000, and was approved (#i9018) by the institutional review board (IRB) of the NYU School of Medicine. The surgical discards from knee replacement surgery were obtained for human chondrocyte isolation and cartilage for histology and Immunohisto-chemistry (IHC) studies. The current study was approved by the Institutional Review Board (IRB) of the NYU School of Medicine, and the ethics committee waived the requirement for informed consent. We have included this information in the Methods section (page no:5; line 7-20). 

- In the Methods, please provide the following information about the mice used in this study: (1) Please state the genetic background of the Ddr1-/- mice; (2) Please provide details of animal welfare (e.g., shelter, food, water, environmental enrichment); (3) Please include the method of euthanasia.

Response: Ddr1-/- and wt mice were maintained on a C57BL6 background, and heterozygote (Ddr1+/-) males and females were bred to produce wild-type (wt) and Ddr1 knockout (Ddr1-/-) littermates for analysis. Mice were fed standard pelleted chow and allowed access to water ad libitum. Mice were euthanized by CO2 inhalation and cervical dislocation, and joint tissues harvested at ages 10-12 weeks for chondrocyte isolation. All animals used in accordance with scientific, humane, and ethical principles and in compliance with regulations approved by the New York University School of Medicine Institutional Animal Care and Use Committee. We have included this information in the Methods section (page no: 6; line 2-10). 

Genotyping was performed from 0.5-cm tail snips collected at 14-21 days postnatally. Genomic DNA was isolated from tail tissues and amplified by PCR using primers used to amplify a 159 (using primer 1 & 2) and 250 base pair (bp) product using (primer 2 & 3) from the mouse Ddr1 gene in wt and mutant respectively were: Primer 1:GCAGCGCATCGCCTTCTATC AND Primer 2: AGACAATCTCGAGATGCTGG AND Primer 3: GTTGCGTTACTCCCGAGATG.

"Funding for this project was provided by the National Institute of Health grant

 AR054817, AR068341, and AR070934. The funders had no role in study design, data

 collection, and analysis, decision to publish, or preparation of the manuscript."

"None"

Response: As per the Journal regulation we have removed the funding information from the acknowledgement section. However, we would like to acknowledge the funding sources in in a funding statement as follow: Funding for this project provided by the National Institute of Health grant AR054817, AR068341, and AR070934. The funders had no role in study design, data collection, and analysis, decision to publish, or preparation of the manuscript."

Response:

We provided the uncropped images of our Western blots in Supporting Information. 

Review Comments to the Author

Reviewer #1: 

In this report, authors would like to prove the interaction of DDR1 and periostin and its signal in chondrocyte. However, the interaction of DDR1 and periostin and DDR1/periostin-MMP13 signal were already proved in several years ago. It is hard to find new finding.

Response: We respectfully disagree with the reviewer’s statement. An extensive literature review indicates that ours is the first report to demonstrate that periostin effects are mediated via the DDR1 signaling pathway. Our description of the DDR1/periostin signal is therefore novel, and we would be pleased to include in our discussion prior literature that demonstrates the DDR1 requirement for periostin signaling that the reviewer can provide.

Here author show that MMP13 was increased when periostin overexpressed chondrocyte cell line, and it is inhibited by DDR1 inhibitor. However, biological meaning of this phenotype is not significant. In OA patient, DDR1 and periostin were not increased by q-PCR and IHC. Actually, periostin and MMP13 increase in OA was reported and recombinant periostin induces DDR1-MMP13 relation was also proved in previously.

Response: We respectfully disagree with this reviewer’s statement. Periostin is strongly elevated in OA cartilage relative to normal cartilage, as shown by qPCR, Western blotting and ELISA [1, 2]. Others have reported elevated expression of periostin in OA synovial fluid and associated with severity [1-8]. In our manuscript we do report that DDR1 is not differentially expressed in OA vs. normal cartilage. However, we point out to the reviewer that lack of upregulation of DDR1 per se does not diminish the “biological significance” of the finding as he/she suggests. First, because its “ligand” periostin is upregulated, and second, because signaling molecules need not be upregulated to become activated. We would like to highlight reviewer 3’s comments here “This is an interesting article with original results”.

Please show us new signaling of periostin-DDR1 or new biological meaning.

Response: In the current study we present evidence that periostin interacts with collagen receptor DDR1 tyrosine kinase receptor and activates the AKT/β-catenin pathway, inducing MMP-13 expression in human and mouse chondrocytes. As noted above, ours is the first report to show that periostin binds to collagen receptor DDR1 and induces MMP-13 expression through this interaction. The “new biological meaning” requested by the reviewer is apparent: this observation elucidates a new signaling mechanism for a periostin, a catabolic ECM molecule implicated in the pathogenesis of OA. Insight into its signaling pathway provides new opportunities for treatment strategies.

Reviewer #2: 

Excellent paper with solid evidences deciphering the periostin pathway in chondrocytes, and showing that the pathway is different from bone pathway.

The discussion is very well constructed, with the question raised by the spontaneous OA developed by ddr-/- mice.

Response: We thank the reviewer for these comments. 

Just minors punctuation errors (pg 10 line 12 .,. , and some "Figure" instead "Fig. ")

Response: We thank the reviewer for highlighting these errors, and corrected them all.

Reviewer #3: 

In a paper previously published in FASEB J in 2015, the authors reported high periostin levels in human and mouse OA cartilage. Gain and loss of function experiments indicated that periostin stimulated MMP13 expression and promoted cartilage degradation. In the present article, they report that periostin induces collagen and proteoglycan degradation by signaling through discoidin domain receptor-1 (DDR1).

They first show that DDR1 mRNA and protein levels are not different between non-lesion and lesion OA, and was cell-associated. In contrast periostin protein was more expressed in OA lesions and was both cell and ECM-associated. Then by performing co-immunoprecipitation, they show in OA chondrocytes and in the C28I2 cell line that DDR1 interacts with periostin. Inhibition of DDR1 with the 7rh molecule blocked the increase in MMP13 levels induced by periostin adenoviral overexpression. Moreover, authors show that periostin activates AKT phosphorylation and increased beta-catenin levels, and that these effects are prevented by inhibition of DDR1. Finally, in chondrocytes from Ddr1 deficient mice, periostin does not increase MMP13 expression and activity further suggesting that periostin signals through DDR1.

This is an interesting article with original results. 

The results are convincing. However, the methods used and the experimental conditions are insufficiently detailed in the materials and methods section and in the figure legends.

Response: We have revised and updated the Methods section to provide detailed experimental conditions (page nos: 5-7).

- Abstract line 16: 2 spaces between DDR1 and is. Several other similar errors throughout the text

Response: We would like to thank the reviewer for highlighting these errors. We edited the manuscript accordingly.

- Introduction: periostin is presented as a TGFβ-inducible protein. TGFβ is known as an anabolic cartilage GF and periostin catabolic in the context of OA. Can authors explain this apparent opposition?

Response: Multiple lines of evidence show that TGF-β signaling acts as a double edge sword in healthy and OA joint tissues [9] . 1). TGF-β signals via multiple pathways. Different TGF-β receptors control different downstream signals: ALK5 binding of TGF-β leads to Smad2/3 signaling which is associated with an anabolic effect on cartilage, whereas ALK1 activation leads to Smad1/5/8 signaling associated with increased expression of MMP-13 and loss of matrix [10]. 2) TGF-β has been shown to exert catabolic effects following a shift in the ALK1 or ALK5 receptor expression with age and/or disease [10]. 3) Inhibition of Smad signaling protects against cartilage injury and osteoarthritis [11]. 4) Activation of TGF-β signaling in mesenchymal progenitor cells of subchondral bone also causes OA-like lesions [11]. 5) The circadian rhythm pathway is dysregulated in OA cartilage [12] and interference with circadian rhythmicity in chondrocytes affects TGF-β signaling, and cartilage homeostasis. 6) Periostin expression is lower in normal relative to OA cartilage. Only under pathological conditions is periostin expression elevated in cartilage. Periostin expression is increased during disease progression due to dysregulated TGFβ signaling, and leads to activation of AKT/ β-catenin signaling and MMP-13 expression. 

- Please indicate whether you used human chondrocytes with several passages or only freshly isolated cells.

Response: We used primary human chondrocytes at passage 1 or 2 in all our experiments. We reported this information in the Methods section of our revised manuscript (page no 6; line 18-21).

- Please indicate in the legend of figure 3 what corresponds to A or B.

Response: We indicated panel A & B in the Figure 3 legend of our revised manuscript.

- Page 11 first sentence: “strongly upregulated” written twice

- Page 11, first paragraph, last sentence (and figure 3). It is necessary to explain why authors used a wnt inhibitor in this experiment. They also have to explain which Wnt inhibitor was used, at which concentration, and discuss the results (MMP13 levels completely dropped). Is this Wnt inhibitor cytotoxic at the concentration used?

Response: We used the Wnt inhibitor CCT031374 hydrobromide (cat. no. 4675/10, R&D Systems, Minneapolis, MN, USA) at 1-50 uM concentrations in our in vitro experiments. CCT031374 was not toxic in human chondrocytes, as we reported in our previous publication [1], where we showed that this Wnt inhibitor blocked periostin-induced MMP-13 expression and enzyme activity. In the current study we used the Wnt inhibitor (10uM) as a positive control in our experiments. 

- Page 11 and Figure 4A: authors should perform densitometric quantification of the bands from several independent experiments and calculate the mean to better analyze DDR1 phosphorylation in response to periostin.

- Page 11 figure 4B: same remark. The authors have to perform densitometric analyses to fully convince the readers.

Response: As requested, we performed quantitation of the Western blots. We thank the reviewer for this suggestion, which we believe significantly strengthened our manuscript.

- Page 11 and page 13: authors should explain the molecular link between Akt and b-catenin. Is this activation of b-catenin independent from the canonical Wnt signaling pathway activated by Wnt growth factors?

 Response: We thank the reviewer for this comment, which gave us the opportunity to better explain the signaling mechanism activated by Postn binding to DDR1. Our data show that periostin – DDR1 interaction results in AKT activation and increased b-catenin levels, as well as upregulation of MMP_13 expression, effects blocked by pharmacological inhibition or genetic deficiency of DDR1. We had previously reported that inhibition of b-catenin signaling abrogates periostin induction of MMP-13 expression. These findings show that periostin – DDR1 signaling is mediated by b-catenin stabilization, and therefore by canonical Wnt signaling. We propose that activation of b-catenin/Wnt signaling is mediated by DDR1 activation of the PI3K-AKT pathway. Active AKT stabilizes b-catenin by two mechanisms, phosphorylation of b-catenin at Ser552 [13] and inhibition of glycogen synthase kinase-3 [14], which phosphorylates β-catenin causing its ubiquitination and degradation. Another potential mechanism could consist of AKT upregulation of canonical Wnt growth factors. However, we could not find any such report in the literature. We briefly discussed these points in our revised manuscript (page no: 15; line 4-11). In addition, we changed the term “Wnt signaling” into “Wnt/ β-catenin signaling” in order to avoid confusion between canonical and non-canonical Wnt signaling. 

6. PLOS authors have the option to publish the peer review history of their article (what does this mean?). If published, this will include your full peer review and any attached files. Response: We agree to publish the peer review history of our article.

We hope that these responses have satisfactorily addressed the reviewers’ comments, and that our revised manuscript will now be acceptable for publication in PLOS ONE. Thank you again for considering our work; we look forward to hearing from you.

Sincerely,

Mukundan Attur PhD

References cited:

1. Attur M, Yang Q, Shimada K, et al. Elevated expression of periostin in human osteoarthritic cartilage and its potential role in matrix degradation via matrix metalloproteinase-13. FASEB J. 2015;29(10):4107-21.

2. Chinzei N, Brophy RH, Duan X, et al. Molecular influence of anterior cruciate ligament tear remnants on chondrocytes: a biologic connection between injury and osteoarthritis. Osteoarthritis Cartilage. 2018;26(4):588-99.

3. Brophy RH, Cai L, Duan X, et al. Proteomic analysis of synovial fluid identifies periostin as a biomarker for anterior cruciate ligament injury. Osteoarthritis Cartilage. 2019.

4. Lourido L, Calamia V, Mateos J, et al. Quantitative proteomic profiling of human articular cartilage degradation in osteoarthritis. J Proteome Res. 2014;13(12):6096-106.

5. Chijimatsu R, Kunugiza Y, Taniyama Y, Nakamura N, Tomita T, Yoshikawa H. Expression and pathological effects of periostin in human osteoarthritis cartilage. BMC Musculoskelet Disord. 2015;16:215.

6. Loeser RF, Olex AL, McNulty MA, et al. Microarray analysis reveals age-related differences in gene expression during the development of osteoarthritis in mice. Arthritis Rheum. 2012;64(3):705-17.

7. Chen KS, Tatarczuch L, Mirams M, Ahmed YA, Pagel CN, Mackie EJ. Periostin expression distinguishes between light and dark hypertrophic chondrocytes. Int J Biochem Cell Biol. 2010;42(6):880-9.

8. Fan B, Liu X, Chen X, et al. Periostin Mediates Condylar Resorption via the NF-kappaB-ADAMTS5 Pathway. Inflammation. 2019.

9. van der Kraan PM. Differential Role of Transforming Growth Factor-beta in an Osteoarthritic or a Healthy Joint. J Bone Metab. 2018;25(2):65-72.

10. Blaney Davidson EN, van Caam AP, van der Kraan PM. Osteoarthritis year in review 2016: biology. Osteoarthritis Cartilage. 2017;25(2):175-80.

11. Wang YJ, Shen M, Wang S, et al. Inhibition of the TGF-beta1/Smad signaling pathway protects against cartilage injury and osteoarthritis in a rat model. Life Sci. 2017;189:106-13.

12. Akagi R, Akatsu Y, Fisch KM, et al. Dysregulated circadian rhythm pathway in human osteoarthritis: NR1D1 and BMAL1 suppression alters TGF-beta signaling in chondrocytes. Osteoarthritis Cartilage. 2017;25(6):943-51.

13. Fang D, Hawke D, Zheng Y, et al. Phosphorylation of beta-catenin by AKT promotes beta-catenin transcriptional activity. J Biol Chem. 2007;282(15):11221-9.

14. Cross DA, Alessi DR, Cohen P, Andjelkovich M, Hemmings BA. Inhibition of glycogen synthase kinase-3 by insulin mediated by protein kinase B. Nature. 1995;378(6559):785-9.

---

## [Decision Letter · Decision Letter 1]

25 Mar 2020

PERIOSTIN INTERACTION WITH DISCOIDIN DOMAIN RECEPTOR-1 (DDR1) PROMOTES CARTILAGE DEGENERATION

PONE-D-19-33141R1

Dear Dr. Attur,

We are pleased to inform you that your manuscript has been judged scientifically suitable for publication and will be formally accepted for publication once it complies with all outstanding technical requirements.

With kind regards,

Dominique Heymann, Ph.D.

Academic Editor

PLOS ONE

Additional Editor Comments (optional):

Reviewers' comments:

Reviewer's Responses to Questions

**Comments to the Author**

1. If the authors have adequately addressed your comments raised in a previous round of review and you feel that this manuscript is now acceptable for publication, you may indicate that here to bypass the “Comments to the Author” section, enter your conflict of interest statement in the “Confidential to Editor” section, and submit your "Accept" recommendation.

Reviewer #3: All comments have been addressed

2. Is the manuscript technically sound, and do the data support the conclusions?

Reviewer #3: Yes

3. Has the statistical analysis been performed appropriately and rigorously? 

Reviewer #3: Yes

4. Have the authors made all data underlying the findings in their manuscript fully available?

Reviewer #3: Yes

5. Is the manuscript presented in an intelligible fashion and written in standard English?

Reviewer #3: Yes

6. Review Comments to the Author

Reviewer #3: Authors have answered the main concerns in a satisfyin way (better description of the materials and methods, densitometric analysis of western blots in particular)

7. PLOS authors have the option to publish the peer review history of their article (what does this mean?). If published, this will include your full peer review and any attached files.

Reviewer #3: No

---

## [Editor Report · Acceptance letter]

27 Mar 2020

PONE-D-19-33141R1 

Periostin Interaction With Discoidin Domain Receptor-1 (DDR1) Promotes Cartilage Degeneration 

Dear Dr. Attur:

I am pleased to inform you that your manuscript has been deemed suitable for publication in PLOS ONE. Congratulations! Your manuscript is now with our production department. 

With kind regards,

on behalf of

Pr. Dominique Heymann 

Academic Editor

PLOS ONE